# Influence of Preoperative Serum Albumin on Acute Kidney Injury after Aortic Surgery for Acute Type A Aortic Dissection: A Retrospective Cohort Study

**DOI:** 10.3390/jcm12041581

**Published:** 2023-02-16

**Authors:** Shijun Xu, Zining Wu, Yongmin Liu, Junming Zhu, Ming Gong, Lizhong Sun, Dong Ran, Hongjia Zhang

**Affiliations:** 1Beijing Aortic Disease Center, Department of Cardiac Surgery, Beijing Anzhen Hospital, Beijing 100029, China; 2Beijing Laboratory for Cardiovascular Precision Medicine, Beijing Institute of Heart Lung and Blood Vessel Diseases, Capital Medical University, Beijing 100029, China; 3Beijing Engineering Research Center of Vascular Prostheses, No.2 Anzhen Street, Beijing 100029, China

**Keywords:** aortic dissection, acute kidney injury, serum albumin

## Abstract

There are relatively few articles on the relationship between serum albumin and acute kidney injury (AKI). Therefore, the objective of this research was to study the relationship between serum albumin and AKI in patients who were undergoing surgery for acute type A aortic dissection. Methods: We retrospectively collected data from 624 patients attending a Chinese hospital between January 2015 and June 2017. The target independent variable was serum albumin measured before surgery after hospital admission, and the dependent variable was AKI, defined in accordance with the Kidney Disease Improving Global Outcomes (KDIGO) criteria. Results: The mean age of these 624 selected patients was 48.5 ± 11.1 years, and almost 73.7% were male. A nonlinear association was detected between serum albumin and AKI; the turning point was 32 g/L. The risk of AKI decreased gradually as the serum albumin level increased up to 32 g/L (adjusted OR = 0.87; 95% CI 0.82–0.92; *p* < 0.001). When the serum albumin level exceeded 32 g/L, the level of serum albumin was not associated with the risk of AKI (OR = 1.01, 95% CI 0.94–1.08; *p* = 0.769). Conclusions: The findings suggest that preoperative serum albumin below 32 g/L was an independent risk factor for AKI in patients undergoing surgery for acute type A aortic dissection. Trial registration: A retrospective cohort study.

## 1. Background

Acute kidney injury (AKI) is not an uncommon postoperative complication of acute type A aortic dissection (ATAAD). Despite improvements in medical management, intensive care unit treatment, and anesthetic and surgical technique, the reported frequency of AKI after surgery for ATAAD remains between 20% and 67%, which is significantly higher than that following other cardiac operations [1,2,3,4,5]. Currently, there are no effective measures for AKI. Therefore, clinicians focus on preventive action and risk factor management [6].

Several studies found that serum albumin may have renal protective effects at cellular and molecular levels [7]. Albumin can effectively promote the reabsorption of interstitial effusion, increase renal flow and urine output, and thus increase circulation volume. In addition, albumin has antioxidant properties, such as scavenging and limiting the production of reactive oxygen species, and providing lysophosphatidic acid with protective effect [8]. Other studies found preoperative hypoalbuminemia may be a potent independent risk factor for AKI after off-pump coronary artery bypass surgery (OPCABG) [9]. However, evidence regarding the relationship between preoperative serum albumin and acute kidney injury following aortic surgery for ATAAD is limited. Due to the complexity of the operation, this kind of procedure is itself an independent risk factor for AKI, including circulatory arrest and longer CPB duration. 

Our center is famous for vascular diseases, especially for the treatment of ATAAD patients. We admit ATAAD patients almost every day and are also very concerned about postoperative AKI. Therefore, we want to determine the relationship between serum albumin and postoperative AKI in ATAAD patients through this study, and also provide strong evidence for precise medical treatment. From our own point of view, this will solve the common but difficulty clinical problems.

## 2. Methods

### 2.1. Study Design and Participants

A retrospective cohort study was carried out at Beijing Anzhen Hospital, which is one of China’s largest cardiac surgery centers and treats thousands of patients with aortic dissection annually from January 2015 to June 2017. The studies involving human participants were reviewed and approved by the human research and development committees of Beijing Anzhen Hospital (approved no. 2018051X), and they complied with the Declaration of Helsinki and principles of good clinical practice. This study is retrospective and no informed consent can be obtained from the patients, and the ethics committee has also approved this protocol. So, individual consent was waived.

A total of 858 patients who underwent aortic surgery for ATAAD within the aforementioned timeframe were admitted to this study. Of these, 23 patients who underwent renal replacement therapy (RRT) before surgery were excluded because the progression of renal dysfunction could not be evaluated. Another 195 patients who had subacute or chronic aortic dissection were excluded. Five patients were also excluded because they died intraoperatively or within 24 h postoperatively. We cannot define AKI without meaningful data. We also excluded patients without sCr or serum albumin values (*n* = 11). Clinical data for the remaining 624 patients were obtained and subsequently analyzed, including demographic data, laboratory data, comorbidities, operative techniques, postoperative morbidity, and mortality. The bromocresol green dye-binding method was used to measure serum albumin concentrations. The reference range of the albumin assay is 40–55 g/L at the hospital. A flowchart for study participant screening and enrolment is shown in Figure 1.

### 2.2. Outcome Variables

The primary endpoint was the incidence of AKI after aortic surgery. Postoperative AKI was diagnosed in accordance with the Kidney Disease Improving Global Outcomes (KDIGO) criteria [10]: increase in sCr ≥ 0.3 mg/dL (at any time within 48 h following operation) or ≥1.5 times by baseline (at any time within 7 days following operation). Urine output was not used, due to potential errors in volume collection and other uncontrollable variables that may arise with retrospective collection.

Other postoperative outcomes, including reoperation for bleeding, dialysis after surgery, mechanical ventilation time, intensive care unit (ICU) time, length of hospital admission, and in-hospital mortality, were also obtained. Indications for postoperative dialysis were significant biochemical abnormalities, anuresis, uremia, and volume overburden.

### 2.3. Judgement of Covariates

Based on our previous studies and other risk factors reported in the literature for AKI after surgery, we selected several covariates, such as hypertension, diabetes mellitus, hemoglobin before surgery, estimated glomerular filtration rate (eGFR), baseline sCr before surgery, preoperative malperfusion syndromes, preoperative renal malperfusion, preoperative blood urea nitrogen (BUN), CPB duration, and intraoperative packed red blood cells (PRBCs). Hemoglobin, baseline sCr, and BUN were obtained from the laboratory testing after the patient was admitted to the hospital but before surgery. 

### 2.4. Surgical Technique

All patients received surgery with median sternotomy and CPB. In patients with ascending aorta replacement, femoral artery intubation was used for CPB. Right axillary artery cannulation was performed for CPB in patients with hemi-arch replacement and total arch replacement and selective cerebral perfusion (SCP) (5–15 mL/(kg·min)). The specific surgical procedures have been described in detail in previous articles [11,12,13,14,15,16,17].

### 2.5. Statistical Analysis

Continuous variables were shown as the mean ± standard deviation or median (quartile), categorical variables were presented as percentages (%) in light of the data dispersion, and the *t*-test was used for comparison if the continuous variables were normally distributed, while the nonparametric Mann-Whitney U test was used if the data were skewed. Categorical variables were compared with the chi-square test. To identify the risk factors for AKI, we used univariate logistic regression analysis. Multiple logistic regression models were used to assess the associations between preoperative serum albumin and AKI after surgery. We established three models: (1) unadjusted; (2) adjusted for demographics, i.e., age; sex; and (3) adjusted for age; sex; BMI; preoperative red blood cell (RBC); preoperative BUN; eGFR; preoperative uric acid; CPB time; operative time; reoperation for bleeding; low cardiac output syndrome; time interval from diagnosis to operation; preoperative malperfusion syndromes; preoperative renal malperfusion. Based on the suggestions of the STROBE statement [18], the results were analyzed from unadjusted or marginally adjusted and completely adjusted data in parallel. Whether the concomitant variable was adjusted was decided according to the following regulation: an adjustment was made if the variable changed the matching odds ratio by at least 10% when added to the model [19]. Smooth curve fitting was performed to detect any nonlinear relationships between the preoperative serum albumin and the risk of AKI in patients who received aortic surgery for ATAAD; this method of application for smooth curve fitting was detailed by Motulsky [20]. Then, the threshold effect between preoperative serum albumin and AKI was analyzed using piecewise regression models, likelihood ratio tests, and bootstrap resampling [21]. We considered it to be statistically significant when a two-tailed *p*-value was less than 0.05. All analyses were implemented with the statistical software package R (http://www. Rproject.org, The R Foundation, accessed on 1 September 2022, Beijng, China) and EmpowerStats (http://www.empowerstats.com, X&Y Solutions, Inc., Boston, MA, USA).

## 3. Results

### 3.1. Characteristics of the Studied Patients

After the exclusion criteria were implemented, 624 consecutive patients with an average age of 48.5 ± 11.1 years were enrolled in this cohort. Of these, 460 (73.7%) were male. The overall incidence rate of AKI was 37.7% (235 patients). Those patients developing AKI were more likely to have advanced age, diabetes, hypertension, lower albumin concentration, lower Hematocrit (%), preoperative malperfusion syndromes, and preoperative renal malperfusion. In-hospital mortality of the AKI group was significantly higher than the Non-AKI group. The characteristics of the 624 study patients are given in Table 1. 

### 3.2. Univariate Analysis of Risk Factors Related to Postoperative AKI in Patients with ATAAD

The results of a univariate analysis are shown in Table 2, which showed that age, male sex, diabetes, hypertension, preoperative acute liver failure, hematocrit, preoperative TP, preoperative serum albumin, uric acid, preoperative BUN, eGFR, sCr, preoperative D-dimer, preoperative TNI, preoperative malperfusion syndromes, preoperative renal malperfusion, intraoperative PRBC transfusion, CPB duration, aortic occlusion time, operative time, minimum rhinopharyngeal temperature, minimum rectal temperature, and reoperation for bleeding were significantly correlated with postoperative AKI.

### 3.3. The Nonlinear Relationship between Preoperative Serum Albumin and AKI after Adjusting for Covariates

After adjusting for these possible factors related to AKI, including age, sex, BMI, preoperative RBC, preoperative BUN, eGFR, preoperative uric acid, CPB time, operative time, reoperation for bleeding, and low cardiac output syndrome, a nonlinear relationship between age and AKI was observed (Figure 2) in smooth curve fitting. Table 3 shows the threshold influence of preoperative serum albumin on the hazard of AKI from piecewise linear regression. In Model I (unadjusted), when preoperative serum albumin was less than the turning point (32 g/L), it was inversely related to the risk of AKI (OR = 0.86, 95% CI: 0.82–0.91; *p* < 0.001). When preoperative serum albumin was more than 32 g/L, it was not related to the risk of AKI (OR = 1.00, 95% CI: 0.94–1.07; *p* = 0.898). In Model II (adjusted for age and sex), when preoperative serum albumin was less than 32 g/L, it was inversely related to the risk of AKI (OR = 0.86, 95% CI: 0.81–0.90; *p* < 0.001). When preoperative serum albumin was more than 32 g/L, it was not related to the risk of AKI (OR = 1.01, 95% CI: 0.95–1.08; *p* = 0.796). In Model III (adjusted for: age; sex; BMI; preoperative RBC; preoperative BUN; eGFR; preoperative uric acid; CPB time; operative time; reoperation for bleeding; low cardiac output syndrome; Time interval from diagnosis to operation; preoperative malperfusion syndromes; preoperative renal malperfusion), when preoperative serum albumin was less than 32 g/L, it was inversely related to the risk of AKI (OR = 0.85, 95% CI: 0.79–0.91; *p* < 0.001). When preoperative serum albumin was more than 32 g/L, it was not related to the risk of AKI (OR = 1.04, 95% CI: 0.95–1.15; *p* = 0.374). (LRT: *p* < 0.05 indicates a nonlinear relationship between preoperative serum albumin and AKI.)

## 4. Discussion

The present study analyzed the results of 624 Chinese patients obtained for a retrospective cohort study where each participant underwent aortic surgery for STAAAD. We found, for the first time, that preoperative serum albumin was conversely related to the hazard of AKI in the presence of less than 32 g/L among Chinese patients even after adjustment for significant confounding factors before surgery and perioperatively. In the event that the preoperative serum albumin increased by 1 g/L, the risk of postoperative AKI decreased by 15%. When preoperative serum albumin exceeded 32 g/L, it was not associated with the risk of postoperative AKI.

Numerous clinical studies have identified preoperative serum albumin as a notable risk factor for AKI among patients undergoing cardiac surgery. In addition, a previous meta-analysis indicated that hypoalbuminemia was an independent hazard of AKI [22]. Despite the strong association between hypoalbuminemia and AKI, a limited number of studies have evaluated the influence of serum albumin levels before surgery on AKI following surgery in patients who received aortic surgery for ATAAD. Kim et al. analyzed 702 patients undergoing surgery on the thoracic aorta with CPB, 352 of whom underwent aortic dissection [23]. The study showed that preoperative albumin <4.0 g/dL (OR = 2.50; CI 1.39–4.50; *p* = 0.008) was an independent risk factor for AKI. Several existing cohort studies have obtained similar results. Lee et al. evaluated the effect of preoperative albumin (<4 g/dL) on postoperative AKI in 1182 patients undergoing OPCABG and observed that hypoalbuminemia was an independent hazard of postoperative AKI after adjustment for confounders by logistic regression and propensity analyses [8]. An additional study assessed the association between preoperative hypoalbuminemia and postoperative adverse events in 5168 patients who underwent coronary artery bypass grafting (CABG) with or without valve surgery [24] and found that patients with hypoalbuminemia (<2.5 g/dL) had a substantial risk of renal failure (OR = 2.0, 95% CI: 1.3–3.2) following surgery after adjustment for confounders with multivariable logistic regression. Moreover, several other studies have shown that preoperative hypoalbuminemia was an independent predictive factor of postoperative AKI in patients undergoing cardiac operation, which further verifies the findings of the present research study [25,26].

Preoperative serum albumin may be causally associated with the progression of AKI following surgery rather than acting merely as a marker for alternative pathophysiological courses. Existing literature shows that serum albumin operates as a renal defense component that influences both cells and molecules. Kaufmann et al. observed that albumin could improve kidney perfusion and glomerular filtration via increased potent renal vasodilation [7]. Iglesias et al. affirmed that albumin could prevent apoptosis of renal tubular cells by scavenging reactive oxygen species and transporting protective lysophosphatidic acid [27]. Furthermore, Dixon et al. discovered that albumin could stimulate the proliferation of renal tubular cells via activation of phosphatidylinositide 3-kinase [28]. All of these findings indicate the significance of albumin in the preservation of the integrity and function of tubular cells.

The present research advances evidence from existing studies in two aspects. First, this investigation adopts the widely accepted AKI definition after cardiac surgery as the outcome (KDIGO criteria), which represents the currently recognized diagnostic criteria. Second, the study cohort comprised consecutive patients with different risks undergoing various aortic surgical procedures, so our study encompasses a broader population. 

In comparison to the literature, the present study included a large number of participants, particularly ATAAD patients undergoing aortic surgery. The data were obtained by trained nurses who were blinded to the study. During data collection, some nurses only collected the preoperative data without knowing outcomes, and others only collected outcome data without knowing preoperative data. In addition, the present study applied KDIGO recommendations for AKI in place of the earlier two guidelines, as the KDIGO recommendations were recently modified and offer clarity and succinctness in clinical application. Thus, smooth curve fitting was introduced to determine nonlinear correlations between preoperative serum albumin and the risks of AKI in patients who were receiving aortic surgery. There was an adjustment for confounding factors, and the axis turning point was 32 g/L. Finally, this study was observational and therefore easily affected by underlying confounders. There was rigorous statistical adjustment to minimize underlying confounding factors and enhance the validity of these consequences.

The findings may have significant clinical implications. It is important to deepen our understanding of hazardous factors related to AKI progression, and our emphasis on preoperative albumin levels (<32 g/L) as a hazardous factor may direct clinical treatment tactics. Early recognition of high-risk patients for postoperative AKI allows clinical application specialists to strictly observe these patients and to implement preventive and therapeutic methods to decrease the occurrence of AKI. Such methods include avoidance of specific medication that can damage the renal system [29], reduction in time allocated to CPB, rapid identification of AKI, and prompt intervention [30]. Indeed, as we can see, a lot of factors influence the development of this complication. It does not mean that in patients with serum albumin over 32 g/L we do not need to avoid any specific medication that can damage the renal system, or limit time allocated to CPB, etc. Furthermore, if the association between preoperative serum albumin and AKI following aortic surgery is definitely causal, serum albumin before surgery may be a changeable risk variable. Lee discovered that in patients with preoperative serum albumin levels below 4.0 g/dL, prompt administration of 20% exogenous albumin before cardiac surgery increases urine output during surgery and decreases the risk of AKI after OPCABG [9]. Whether exogenous albumin supplementation is safe and beneficial for use in deep hypothermic circulatory arrest cases requires further verification. Moreover, the results of this research will be helpful for future studies to establish diagnostic or predictive models for AKI [31].

The present study had a number of limitations. First, the patients in our study were undergoing aortic surgery for ATAAD. Therefore, there are some deficiencies in the extrapolation and universality of the research. Second, there were several exclusion factors that removed the data of specific participants, namely, those who required RRT for kidney injury preoperatively and patients who died intraoperatively or within 24 h of the operation, which means that the results of this study are only applicable to these patients. Third, the majority of patients in the sample were male. More caution is needed when applying the findings to female patients. Fourth, the study was observational and nonrandomized. Fifth, albumin can change over time, and only using a single time point for albumin may lead to a difference in results. We chose the results that were closest to the time before the operation and tried our best to eliminate the differences in the results caused by changes in albumin. Furthermore, intravenous albumin before surgery may also confound the current finding. However, no patients in our center received intravenous albumin before surgery. Therefore, more caution is needed when applying the findings to patients who have already been received with albumin.

Although numerous variables regarded as affecting AKI following aortic surgery were taken into consideration, the influences of hidden or non-investigated risk factors cannot be completely excluded. The method of albumin measurement (bromocresol green) has been shown to be inferior to the bromocresol purple method in some studies, including those with inflammation or acute-phase globulins. Finally, given the observational nature of the research, we could not obtain a causal correlation between preoperative serum albumin and the risk of AKI following aortic surgery.

## 5. Conclusions

These findings suggest a nonlinear relationship between preoperative serum albumin and AKI in patients who underwent aortic surgery for ATAAD. Serum albumin less than 32 g/L before surgery was independently associated with an increased risk of AKI after aortic surgery for ATAAD. We need to design new studies to understand the molecular mechanisms of this connection so that preventative therapies can be developed accordingly.

## Figures and Tables

**Figure 1 jcm-12-01581-f001:**
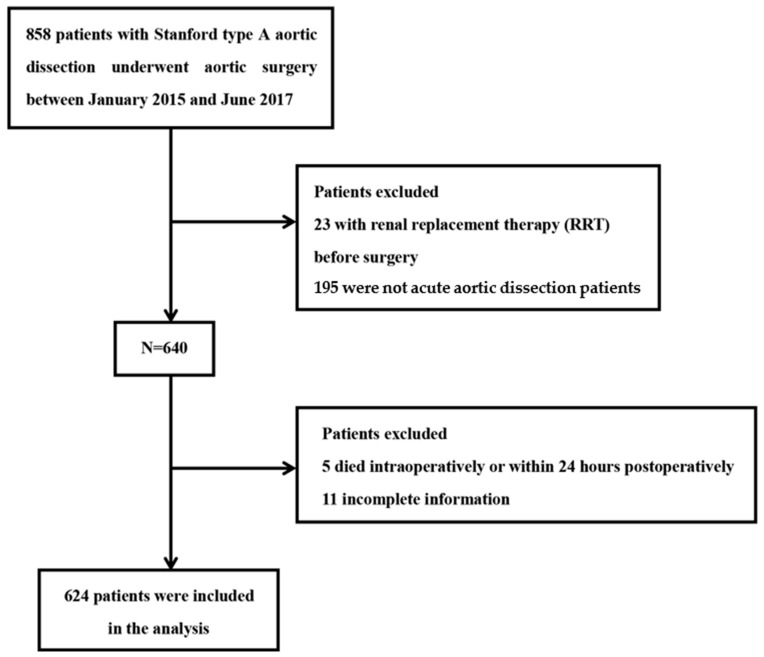
Flow diagram of the screening and enrolment of study patients. After exclusion criteria were applied, 624 consecutive patients were included in this cohort.

**Figure 2 jcm-12-01581-f002:**
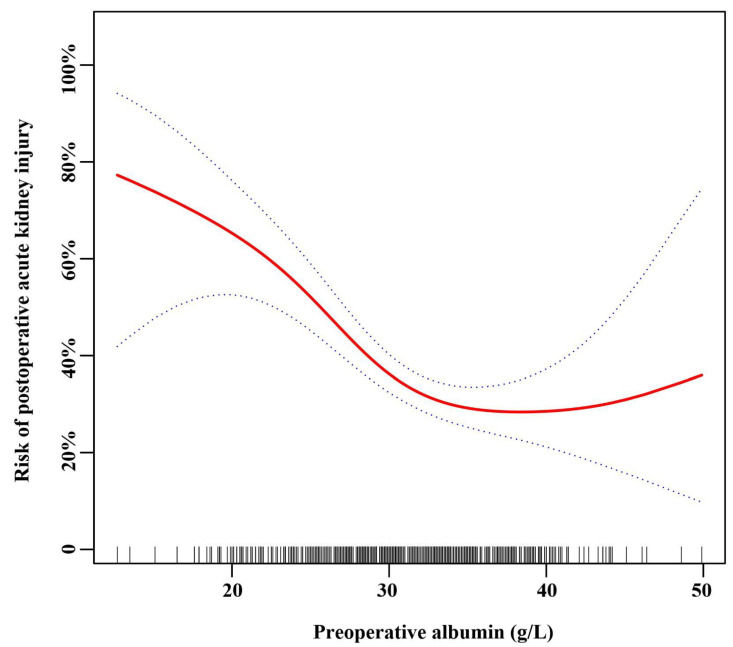
Spline smoothing was performed using a GAM (generalized additive model) to explore the association between preoperative serum albumin and postoperative acute kidney injury after adjusting for: age; sex; body mass index; preoperative red blood cells; preoperative blood urea nitrogen; estimated glomerular filtration rate; preoperative uric acid; cardiopulmonary bypass time; operative time; reoperation for bleeding; low cardiac output syndrome; time interval from diagnosis to operation; preoperative malperfusion syndromes; preoperative renal malperfusion. The red line indicates the estimated risk of AKI, and the dotted lines represent the pointwise 95% confidence interval. The vertical lines of the *X*-axis indicate the distribution of the single observation.

**Table 1 jcm-12-01581-t001:** Characteristics of study patients at baseline.

Variables	Non-AKI	AKI	*p*-Value
N	389	235	
Age (y)	47.0 ± 10.6	50.9 ± 11.4	<0.001
Gender			0.008
female	88 (22.6%)	76 (32.3%)	
male	301 (77.4%)	159 (67.7%)	
BMI (kg/m^2^)	26.1 ± 4.0	26.0 ± 3.8	0.691
Smoking history	130 (33.4%)	79 (33.6%)	0.960
Chronic obstructive pulmonary disease	1 (0.3%)	2 (0.9%)	0.327
Diabetes	8 (2.1%)	13 (5.5%)	0.025
Hypertension	191 (49.1%)	140 (59.6%)	0.011
Previous cerebral infarction	9 (2.3%)	4 (1.7%)	0.606
Previous cardiovascular disease	23 (5.9%)	17 (7.2%)	0.752
Congestive heart failure	2 (0.5%)	2 (0.9%)	0.613
Preoperative acute liver failure	23 (6.1%)	32 (13.6%)	0.002
Preoperative shock	2 (0.5%)	2 (0.9%)	0.613
History of CABG	0 (0.0%)	2 (0.9%)	0.981
History of aortic surgery	4 (1.0%)	1 (0.4%)	0.428
History of valve surgery	2 (0.5%)	2 (0.9%)	0.613
Preoperative RBC	3.5 ± 0.8	3.5 ± 0.8	0.383
Preoperative Hemoglobin (g/L)	111.5 ± 23.1	108.5 ± 24.2	0.119
Hematocrit (%)	38.5 ± 4.9	37.5 ± 5.3	0.020
Preoperative TP (g/L)	51.3 ± 8.6	47.8 ± 10.7	<0.001
Preoperative albumin (g/L)	32.3 ± 5.2	29.5 ± 6.2	<0.001
Preoperative Uric acid (μmol/L)	332.7 ± 114.2	372.6 ± 147.3	<0.001
Preoperative BUN (mmol/L)	9.2 ± 3.4	10.8 ± 4.7	<0.001
Preoperative sCr (mg/dL)	1.2 ± 0.5	1.5 ± 1.0	<0.001
eGFR mL/(min·1.73 m^2^)	76.0 ± 25.3	67.0 ± 35.9	<0.001
Preoperative D-dimer (ng/mL)	(1010.2–2736.2)	(1070.0–3704.0)	<0.001
Preoperative TNI (ng/mL)	4.0 (2.2–7.0)	5.7 (3.5–13.1)	<0.001
Mean arterial pressure at admission (mmHg)	89.4 ± 13.9	87.8 ± 13.9	0.179
LVEF (%)	61.8 ± 5.7	61.6 ± 6.5	0.818
Time interval from diagnosis to operation(d)	1.0 (0.8–2.5)	1.0 (0.6–2.0)	0.270
Preoperative malperfusion syndromes	222 (57.1%)	179 (76.2%)	<0.001
Preoperative renal malperfusion	29 (7.5%)	51 (21.7%)	<0.001
Preoperative cardiac tamponade	49 (12.6%)	24 (10.2%)	0.370
Intraoperative PRBCs	64 (16.5%)	48 (20.4%)	0.211
Aortic root surgery			0.694
no	2 (0.5%)	2 (0.9%)	
Bentall	175 (45.0%)	96 (40.9%)	
Wheat	3 (0.8%)	1 (0.4%)	
David	1 (0.3%)	0 (0.0%)	
Aortic arch surgery			0.294
no	17 (4.4%)	5 (2.1%)	
Hemi-arch replacement	37 (9.5%)	20 (8.5%)	
Total arch replacement	335 (86.1%)	210 (89.4%)	
Descending aortic surgery			0.206
no	55 (14.1%)	25 (10.6%)	
Elephant trunk stent	334 (85.9%)	210 (89.4%)	
Combined with other cardiac surgery			
CABG	18 (4.6%)	14 (6.0%)	0.467
MVR	4 (1.0%)	4 (1.7%)	0.473
TVP	3 (0.8%)	1 (0.4%)	0.605
Aortic bypass surgery	31 (8.0%)	27 (11.5%)	0.144
CPB time (min)	200.8 ± 54.7	220.4 ± 56.5	<0.001
Aortic cross clamp time (min)	112.9 ± 33.6	125.6 ± 38.8	<0.001
Circulatory arrest	362 (93.1%)	227 (96.6%)	0.063
Circulatory arrest time (min)	22.6 ± 10.6	24.0 ± 10.7	0.114
Operative time (h)	7.5 ± 1.9	8.3 ± 1.9	<0.001
Minimum nasopharyngeal temperature (°C)	24.0 ± 1.9	23.4 ± 1.9	<0.001
Minimum rectal temperature (°C)	25.7 ± 2.3	25.2 ± 2.0	0.005
Postoperative dialysis	0 (0.0%)	95 (40.4%)	<0.001
Reoperation for bleeding	9 (2.3%)	20 (8.5%)	<0.001
Low cardiac output syndrome	5 (1.3%)	8 (3.4%)	0.084
Mechanical ventilation time (h)	38.0 (22.0–82.0)	131.0 (62.0–252.5)	<0.001
Length of ICU stay (d)	1.7 (1.0–2.7)	4.3 (2.2–7.9)	<0.001
Length of in hospital stay (d)	11.0 (9.0–15.0)	14.0 (10.0–19.0)	<0.001
In-hospital mortality	14 (3.6%)	46 (19.6%)	<0.001

The results are expressed as *n* (%) or mean ± SD or median (IQR). AKI = acute kidney injury; BMI = body mass index; CPB = cardiopulmonary bypass; ICU = intensive care unit; PRBCs = packed red blood cells; BUN = blood urea nitrogen; sCr = serum creatinine; SD = standard deviation; IQR = interquartile range. TP = total protein; CABG = coronary artery bypass grafting; RBC = red blood cell; eGFR = estimated glomerular filtration rate; LVEF = left ventricular ejection fraction; TEVAR = thoracic endovascular aneurysm repair; MVR = mitral valve replacement; TVP = tricuspid valvuloplasty.

**Table 2 jcm-12-01581-t002:** Univariate analysis of risk factors associated with postoperative acute kidney injury in patients with acute type A aortic dissection.

Variable	Statistics	OR (95% CI)	*p*-Value
Age (y)	48.5 ± 11.1	1.03 (1.02, 1.05)	<0.001
Gender			
female	164 (26.3%)	1.0	
male	460 (73.7%)	0.61 (0.43, 0.88)	0.008
BMI (kg/m^2^)	26.1 ± 3.9	0.99 (0.95, 1.03)	0.691
Smoking history	209 (33.5%)	1.01 (0.72, 1.42)	0.960
Chronic obstructive pulmonary disease	3 (0.5%)	3.33 (0.30, 36.93)	0.327
Diabetes	21 (3.4%)	2.79 (1.14, 6.83)	0.025
Hypertension	331 (53.0%)	1.53 (1.10, 2.12)	0.011
Previous cerebral infarction	13 (2.1%)	0.73 (0.22, 2.40)	0.606
Previous cardiovascular disease	40 (6.4%)	1.11 (0.58, 2.14)	0.752
Congestive heart failure	4 (0.8%)	0.6 (0.06, 5.78)	0.656
Preoperative acute liver failure	55 (9.0%)	2.44 (1.39, 4.28)	0.002
Preoperative shock	5 (0.8%)	2.50 (0.42, 15.10)	0.317
History of CABG	2 (0.3%)	- §	0.981
History of aortic surgery	5 (0.8%)	0.41 (0.05, 3.70)	0.428
History of valve surgery	4 (0.6%)	1.66 (0.23, 11.87)	0.613
Preoperative RBC	3.5 ± 0.8	0.91 (0.75, 1.12)	0.383
Preoperative Hemoglobin (g/L)	110 ± 24	0.99 (0.99, 1.00)	0.119
Hematocrit (%)	38.1 ± 5.1	0.96 (0.93, 0.99)	0.020
Preoperative TP (g/L)	50.0 ± 9.6	0.96 (0.94, 0.98)	<0.001
Preoperative albumin (g/L)	31.2 ± 5.7	0.92 (0.89, 0.94)	<0.001
Preoperative Uric acid (μmol/L)	347.7 ± 129.0	1.00 (1.00, 1.00)	<0.001
Preoperative BUN (mmol/L)	9.8 ± 4.0	1.11 (1.06, 1.15)	<0.001
Preoperative sCr (mg/dL)	1.3 ± 0.7	1.01 (1.00, 1.01)	<0.001
eGFR mL/(min·1.73 m^2^)	72.6 ± 30.1	0.99 (0.98, 1.00)	<0.001
Preoperative D-dimer (ng/mL)	1973 (1031.0–2959.8)	1.00 (1.00, 1.00)	<0.001
Preoperative TNI (ng/mL)	4.7 (2.5–8.6)	1.03 (1.01, 1.04)	<0.001
Mean arterial pressure at admission (mmHg)	88.8 ± 13.9	0.99 (0.98, 1.00)	0.179
LVEF (%)	61.9 ± 6.0	1.00 (0.97, 1.03)	0.818
Time interval from diagnosis to operation(d)	1.93 ± 2.12	0.95 (0.87, 1.04)	0.271
Preoperative malperfusion syndromes	401 (64.26%)	2.40 (1.68, 3.45)	<0.001
Preoperative renal malperfusion	80 (12.82%)	3.44 (2.11, 5.61)	<0.001
Preoperative cardiac tamponade	73 (11.7%)	0.79 (0.47, 1.32)	0.370
Intraoperative PRBCs	112 (18.0%)	1.30 (0.86, 1.97)	0.211
Aortic root surgery			
no	348 (55.8%)	1.0	
Bentall	271 (43.4%)	0.83 (0.60, 1.16)	0.282
Wheat	4 (0.6%)	0.51 (0.05, 4.93)	0.558
David	1 (0.2%)	- §	0.980
Aortic arch surgery			
no	22 (3.5%)	1.0	
Hemi-arch replacement	57 (9.1%)	1.84 (0.59, 5.72)	0.294
Total arch replacement	545 (87.3%)	2.13 (0.77, 5.86)	0.143
Descending aortic surgery			
no	80 (12.8%)	1.0	
Elephant trunk stent	544 (87.2%)	1.38 (0.84, 2.29)	0.206
Combined with other cardiac surgery			
CABG	32 (5.1%)	1.31 (0.64, 2.68)	0.467
MVR	8 (1.3%)	1.67 (0.41, 6.73)	0.473
TVP	4 (0.6%)	0.55 (0.06, 5.32)	0.605
Aortic bypass surgery	58 (9.3%)	1.50 (0.87, 2.58)	0.144
CPB time (min)	208.2 ± 56.1	1.01 (1.00, 1.01)	<0.001
Aortic cross clamp time (min)	117.7 ± 36.2	1.01 (1.01, 1.01)	<0.001
Circulatory arrest	589 (94.39%)	2.12 (0.95, 4.74)	0.063
Circulatory arrest time (min)	23.1 ± 10.7	1.01 (1.00, 1.03)	0.114
Operative time (h)	7.8 ± 1.9	1.26 (1.14, 1.39)	<0.001
Minimum nasopharyngeal temperature (°C)	23.8 ± 1.9	0.84 (0.76, 0.92)	<0.001
Minimum rectal temperature (°C)	25.5 ± 2.2	0.89 (0.83, 0.97)	0.005
Reoperation for bleeding	29 (4.7%)	3.93 (1.76, 8.78)	<0.001
Low cardiac output syndrome	13 (2.1%)	2.71 (0.87, 8.37)	0.084

The bold value indicates significance at *p* < 0.05. § = the result failed due to small sample size. The results are expressed as *n* (%) or mean ± SD or median (IQR). OR = odds ratio; CI = confidence interval; AKI = acute kidney injury; BMI = body mass index; CPB = cardiopulmonary bypass; PRBCs = packed red blood cells; BUN = blood urea nitrogen; TP = total protein; sCr = serum creatinine; SD = standard deviation; IQR = interquartile range; CABG = coronary artery bypass grafting; RBC = red blood cell; eGFR = estimated glomerular filtration rate; LVEF = left ventricular ejection fraction; TEVAR = thoracic endovascular aneurysm repair; MVR = mitral valve replacement; TVP = tricuspid valvuloplasty.

**Table 3 jcm-12-01581-t003:** Threshold effect of preoperative albumin on postoperative AKI in patients with acute type A aortic dissection.

	Model I OR (95% CI)	*p*-Value	Model II OR (95% CI)	*p*-Value	Model III OR (95% CI)	*p*-Value
Model A						
One-line slope	0.92 (0.89, 0.94)	<0.001	0.92 (0.89, 0.95)	<0.001	0.92 (0.88, 0.96)	<0.001
Model B						
Turning point (K)	32 (g/L)		32 (g/L)		32 (g/L)	
<K slope 1	0.86 (0.82, 0.91)	<0.001	0.86 (0.81, 0.90)	<0.001	0.85 (0.79, 0.91)	<0.001
>K slope 2	1.00 (0.94, 1.07)	0.898	1.01 (0.95, 1.08)	0.796	1.04 (0.95, 1.15)	0.371
LRT test		0.002		0.002		0.004

OR = odds ratio; CI = confidence interval; AKI = acute kidney injury; BMI = body mass index; BUN = blood urea nitrogen; CPB = cardiopulmonary bypass; eGFR = estimated glomerular filtration rate; LRT = log likelihood ratio; RBC = red blood cell; Model I: unadjusted; Model II: adjusted for age; gender; Model III: adjusted for age; sex; BMI; preoperative RBC; preoperative BUN; eGFR; preoperative uric acid; CPB time; operative time; reoperation for bleeding; low cardiac output syndrome; time interval from diagnosis to operation; preoperative malperfusion syndromes; preoperative renal malperfusion.

## Data Availability

The datasets generated or analyzed during this study are available from the corresponding author on reasonable request.

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
