# Peer review of "Influence of Preoperative Serum Albumin on Acute Kidney Injury after Aortic Surgery for Acute Type A Aortic Dissection: A Retrospective Cohort Study"

_jcm, 2023, doi:10.3390/jcm12041581_

Round 1
Reviewer 1 Report
Dear authors,
I have studied with great interest the manuscript «Influence of preoperative serum albumin on acute kidney injury after aortic surgery for acute type A aortic dissection: a Retrospective cohort Study».
Acute type A aortic dissections are one the most common forms of aortic lesion and further studies are needed on acute kidney injury prediction due to its high prevalence in those patients. The authors concluded that preoperative serum albumin below 32 g/L was an independent risk factor for AKI in patients undergoing surgery for acute type A aortic dissection.
The manuscript is clearly exposed and well written. The topic is original. The references are appropriate. The figures correspond to the description in the text, are well designed and reflect important information.
But I have some comments:
Initially, the heterogeneous groups of patients were included in the study; the AKI group had more severe comorbidities and a higher number of postoperative complications, and as a consequence, a higher frequency of AKI. Of course, it is great that the authors confirmed the role of hypoalbuminemia in the development of AKI, but we should not be guided by only the level of albumin to anticipate AKI development. Indeed, as we can see, a lot of factors influence the development of this complication. It does not mean that in patients with serum albumin over 32 g/L we should not avoid any specific medication that can damage the renal system, or limit time allocated to CPB, etc.
I suppose it should be discussed that, for sure, it is important results but we have to interpret them in addition to clinical data.
Generally, I think that this is a very worthy work. I express my gratitude to the authors for their work and my great pleasure in reading their results.
Reviewer 2 Report
Review of article: ‘Influence of preoperative serum albumin on acute kidney injury after aortic surgery for acute type A aortic dissection: a Retrospective cohort study- J Clin Med
Xu et al. analyze the role of preoperative serum albumin in the incidence of acute kidney injury after repair of acute type A aortic dissection (ATAAD). They found that a low value of preoperative serum albumin was predictive of this complication after ATAAD repair. The paper analyzed a large cohort of patients operated certainly in an important Chinese referral center for aortic pathology. Nevertheless, the paper has, in my view, some important limitations besides those recognized by the authors.
1. From their results it appears that there are many other significant risk factors for postoperative acute kidney failure. So, why concentrate only on serum albumin? How can be possible to separate the role of all such factors in causing this complication? Does serum albumin be considered not more than a co-factor, albeit important?
2. There is no mention of cases of preoperative malperfusion syndromes. How was this diagnosed? How many patients had preoperative renal malperfusion and how this influenced the results?
3. The issue of malperfusion is very important also in terms of time interval from diagnosis to operation. Did they calculate such interval? Especially in patients with malperfusion, cardiac tamponade or clinical instability delays in repair can adversely influence prognosis and, in this case, also onset of possible complications such as kidney failure.
4. Based on such considerations also the clinical implications of this study should be modified accordingly.
5. I believe that all the previous issues should be adequately addressed and commented.
Round 2
Reviewer 2 Report
The authors have included some new data but substantially my judgement on the paper remains the same because the major issue underlined, the clinicl implicatiions, has not been enough explained.